# Dissipation and Dietary Risk Assessment of Thiacloprid and Tolfenpyrad in Tea in China

Weitao Wang [1,†], Hongping Chen [2,3,†], Di Gao [1], Jiahuan Long [1], Hui Long [1] and Ying Zhang [1,*]

1    Institute of Plant Protection, Guizhou Academy of Agricultural Sciences, Guiyang 550006, China
2    Key Laboratory of Tea Quality and Safety Control, Ministry of Agriculture and Rural Affairs, Hangzhou 310008, China
3    Tea Research Institute, Chinese Academy of Agricultural Sciences, Hangzhou 310008, China
*    Correspondence: zhangying201201@126.com
†    These authors contributed equally to this work.

**Abstract:** Pesticides are widely used to maintain tea yields. For achieving satisfactory effects on pests, multiple pesticides in a low application dose have been the trend at present. In this study, we investigated the dissipation and dietary risk assessment of thiacloprid and tolfenpyrad. A method for the determination of thiacloprid and tolfenpyrad was developed. The recoveries ranged from 73% to 105% with standard deviations between 0.7% and 8.3%. Limits of quantification were 0.01 mg/kg for both pesticides. Field trials were conducted in China in 2021. The half-lives were in ranges of 3.22 to 8.45 days for thiacloprid and 2.91 to 10.50 days for tolfenpyrad. The terminal residues were in the range of 0.04–2.55 mg/kg for thiacloprid and <0.01–4.00 mg/kg for tolfenpyrad, respectively. Finally, a dietary risk assessment was conducted representing the acceptable risk of the two pesticides, which of ratios were all less than 100%. The safe pre-harvest interval of 14 days was suggested. This study can serve as a guide for the rational application of thiacloprid and tolfenpyrad in tea, which also ensures the safety of human health.

**Keywords:** thiacloprid; tolfenpyrad; tea; dietary risk assessment

## 1. Introduction

As one kind of evergreen perennial plants, green tea (*Camellia sinensis* L.) belongs to the angiosperm family of the vegetation classification system, which is planted in tropical and subtropical regions [1,2]. Tea has been the second most popular non-alcoholic beverage next to water consumed all over the world because of its pleasant aroma and health benefits [3]. For instance, tea polyphenols with great antioxidant properties can protect against several diseases (e.g., cancer, cardiovascular disease, and liver disease), as well as the aging process [4–7]. Additionally, this economic crop can be applied to the development of the food industry. For example, tea catechins, such as epicatechin and epigallocatechin gallate [8,9], can inhibit the formation of hazardous substances in some thermally-processed foods, which enables sellers to produce healthier foods. Thus, the tea industry has been continuously expanded to meet the needs of tea in the world. However, tea leaves can be damaged during growing seasons by various pests. Tea thrips (*Scirtothrips dorsalis*) and tea mosquito bugs (*Helopeltis fasciaticollis*) can cause punctures and pale brown dots in leaf regions, respectively, which can lead to severe yield loss [10]. As a simple way with low costs, pesticides are commonly used to improve the yield and quality of tea [11]. However, at the same time, they may pose a risk to humans due to their toxicity [12]. Therefore, it should be determined how to use pesticides rationally so that the toxic effects on human health can be minimized.

Thiacloprid [(Z)-3-(6-chloro-3-pyridylmethyl)-1,3-thiazolidin-2-ylidenecyanamide] (Figure 1a), a type of neonicotinoid insecticide, is highly selective in performing and has excellent systemic activity against sucking and biting insects, such as whiteflies and

aphids [13]. As a potent agonist of insect nicotinic acetylcholine receptors, thiacloprid can bind to γ-aminobutiric acid (GABA) receptors at the postsynaptic membrane. Additionally, it is able to disrupt signal transduction in the insect's central nervous system [14–16]. As for tolfenpyrad (Figure 1b), it is a pyrazole insecticide possessing a pyrazole-carboxamide structure, which is widely used to control pests including Diptera, Lepidoptera, and Hemiptera species [17]. Tolfenpyrad is especially effective against pests resisting pesticides, such as organophosphates and carbamates, whose mechanism of its action is inhibiting complex I in the respiratory electron transfer chain of mitochondria [18]. These two pesticides may produce reproductive and genetic toxicity to humans, and even some teratogenic effects when ingested in higher doses [19–21]. Even though these two pesticides are used to control pests to ensure sufficient production of tea, their residues pollute the surrounding organisms in the food chain, thereby condensing the toxicity of the pesticides, that humans may consume later. In addition, the residues will remain on the final products. All of these can potentially increase the risk to human health. Thus, it is necessary to know the dissipation of thiacloprid and tolfenpyrad so that we can use them rationally and scientifically for tea, which can also maintain the safety of tea for humans.

**Figure 1.** Chemical structures of (**a**) thiacloprid and (**b**) tolfenpyrad.

To date, several studies have just reported either thiacloprid or tolfenpyrad applied to tea. There have been some studies reporting not only the methods of extraction, as well as the residue levels, but also the dissipation and potential risk of neonicotinoids (including thiacloprid) [22–25]. Nevertheless, as far as we know, little work has been conducted on the dietary risks of thiacloprid in tea in China [26,27]. As for tolfenpyrad, although some studies have reported the dietary risk, the samples they collected were just from three places [28,29], and the results may not be generally applicable in China. However, the diversities of climate conditions (e.g., rainfall, temperature, and soil) are so significant that they can cause the variation of residue levels in different regions of China [30]. For this reason, it is extremely important to carry out field trials in different locations. Additionally, compared with a single pesticide, the residue characteristics and dietary risk of multiple pesticides are different [31]. Therefore, the dissipation and the dietary risk of such a new formula consisting of thiacloprid and tolfenpyrad should be clarified so that farmers can use it rationally to produce safe tea products with acceptable risks to consumers.

To evaluate the risk of the combined use of thiacloprid and tolfenpyrad in tea, in this study, we developed a quick method to simultaneously determine thiacloprid and tolfenpyrad in tea samples followed by testing the precision and accuracy of the method. Then field trials were carried out to detect pesticide residues in tea collected at different times from ten locations in China. Finally, we conducted a dietary risk assessment of thiacloprid and tolfenpyrad. Our work not only provides evidence for the safe use of these pesticides in tea but also forms the basis for the recommendations for dietary safety in China.

## 2. Materials and Methods

### 2.1. Mateirals and Reagents

Thiacloprid ($C_{10}H_9ClN_4S$; CAS No. 111988-49-9; 99.4% purity) was purchased from Shanghai Pesticide Research Institute Co., Ltd. (Shanghai, China). Tolfenpyrad ($C_{21}H_{22}ClN_3O_2$; CAS No. 129558-76-5; purity 99.3%) was purchased from Beijing North Weiye Measurement Technology Research Institute (Beijing, China). A 30% concentrated thiacloprid–tolfenpyrad suspension (20% thiacloprid and 10%tolfenpyrad) was obtained from Shanghai Shennong Pesticide Co., Ltd. (Shanghai, China). Acetonitrile was of analytical grade and purchased from Fisher (Marshalltown, IA, US). Formic acid (MS grade), sodium chloride (NaCl, analytical grade), and anhydrous sodium sulfate ($MgSO_4$, analytical grade) were purchased from Beijing Mairuida Technology Co., Ltd. (Beijing, China). High-purity water was prepared in the laboratory. Syringes (2 mL) were purchased from Jiangsu Zhiyu Medical Equipment Co., Ltd. (Jiangsu, China), and 0.22-μm nylon syringe filters were purchased from Tianjin Jinteng Experimental Equipment Co., Ltd. (Tianjin, China).

### 2.2. Field Trials

Field trials with the 30% thiacloprid–tolfenpyrad concentrated suspension were conducted in September 2021 at 10 locations in China as follows: Hefei City, Anhui Province; Qingyuan City, Guangdong Province; Nanning City, Guangxi Province; Guiyang City, Guizhou Province; Enshi Tujia and Miao Autonomous Prefecture, Hubei Province; Changsha City, Hunan Province; Ningbo City, Zhejiang Province; Tai'an City, Shandong Province; Fu'an City, Fujian Province; and Kunming City, Yunnan Province. The trials were carried out according to the Guidelines on Pesticide Residue Trials in China (NY/T 788-2018). Every experiment site was divided into two areas with 100 $m^2$ each and three replicates for each treatment, which were separated from each other by protective signs.

The experimental area was sprayed with the suspension at 4 mL/100 $m^2$ and samples were collected at different intervals (0 (2 h), 7, 14, 21, and 28 days). A total of 3 kg of fresh tea leaf samples were randomly collected from at least 12 sampling points in the area. All of these samples were transported to the laboratory within 8 h and stored in a chamber at −18 °C. A half part of the samples was produced as dried green tea leaves (dry tea) based on the local process of tea production followed by storing in the laboratory under the same conditions as the samples of fresh green tea leaves (fresh tea).

### 2.3. Preparation of the Tea Samples

A total of 2 g of dry or fresh tea samples were weighed on a 50 mL centrifuge tube. Then, 20 mL acetonitrile was added and vortexed for 10 min. After adding 3.0 g NaCl and 3.0 g $MgSO_4$, the mixture was vortexed for 5 min followed by centrifugation for 3 min at 4000 revolutions per min in a model CL5 type centrifuge. The supernatant was injected through a 0.22 μm nylon syringe filter and each filtrate was transferred into a glass vial for subsequent analysis by ultra-high-performance liquid chromatography coupled with tandem mass spectrometry (UHPLC–MS/MS).

### 2.4. UHPLC–MS/MS Analysis

Pesticide residues in the samples and working solutions were analyzed by UHPLC–MS/MS system equipped with an electrospray ionization (ESI) source (1290-6470, Agilent, CA, USA). The analytical column (RRHD Eclipse Plus C18 3.0 × 50 mm, 1.8 μm; Agilent) was used for UHPLC analysis and was maintained at 40 °C. The injection volume was 1 μL. The isocratic elution was performed by using acetonitrile (75%) and 0.1% formic acid (25%) in water at a flow rate of 0.4 mL/min for 2.5 min.

For thiacloprid and tolfenpyrad, MS was performed with positive electrospray ionization ($ESI^+$) in the multiple reaction monitoring (MRM) mode. All ion source and ion optic parameters were optimized as follows: gas temperature 300 °C; gas flow 5.0 L/min; sheath gas temperature 250 °C; sheath gas flow 10.0 L/min; and capillary 3500 V. For thiacloprid, a quantitative transition of 252.8 → 125.9 $m/z$ (collision energy, 21 eV) and a qualitative

transition of 252.8 → 186.1 *m/z* (collision energy, 13 eV) were applied with a fragmentor energy of 139 eV, whose retention time was 0.6 min. For tolfenpyrad, a quantitative transition of 384.2 → 197.1 *m/z* (collision energy, 31 eV) and a qualitative transition of 384.2 → 125.1 *m/z* (collision energy, 35 eV) were applied with a fragmentor energy of 137 eV, whose retention time was 1.9 min. All data were analyzed with Agilent MassHunter Quantitative Analysis 10.0, and the representative chromatograms of the two pesticides were shown in Figure 2.

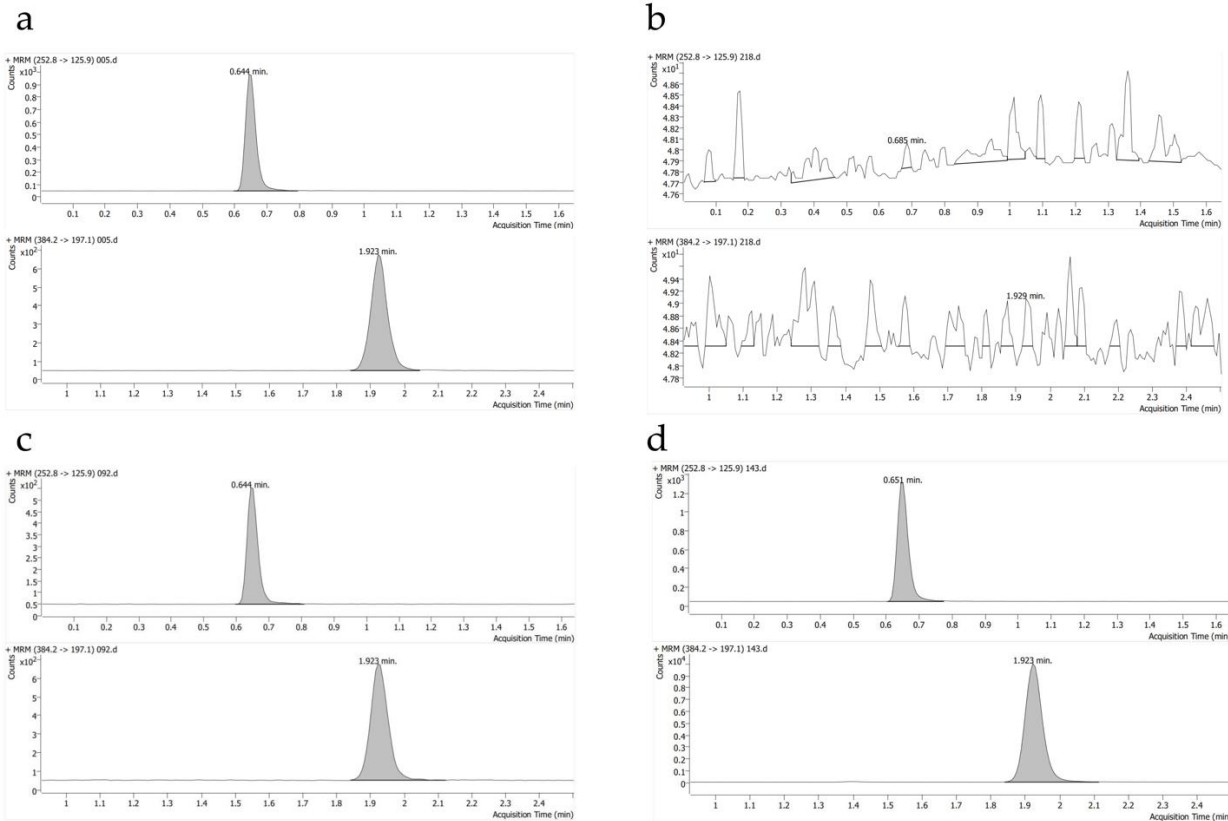

**Figure 2.** The representative UHPLC–MS/MS chromatograms of thiacloprid and tolfenpyrad. (**a**) solvent standard (0.005 mg/L); (**b**) dry tea blank sample; (**c**) dry tea spiked sample (0.05 mg/kg); (**d**) dry tea sample of Anhui province collected on 0 d (2 h).

### 2.5. Method Validation

Complying with the Guidance Document on Pesticide Analytical Methods for Risk Assessment and Post-approval Control and Monitoring Purposes (SANTE/2021/12830) and NY/T 788-2018, the method was validated from the four parameters: sensitivities, recoveries, precisions, and specificities.

To determine the linearity, the standard solutions of thiacloprid and tolfenpyrad were analyzed at 0.001, 0.005, 0.01, 0.05, 0.5, 1.0, and 1.2 mg/L with three replicates. The following parameters were calculated: slope, intercept, and coefficient of determination ($R^2$) for every fitted curve. The matrix effect (ME) was measured by comparing the slope in matrix ($S_1$) and in solvent ($S_2$), according to Equation (1):

$$ME = [(S_1/S_2) - 1] \times 100\% \tag{1}$$

Based on the Maximum Residue Levels (MRLs) of thiacloprid and tolfenpyrad set by China (GB 2763-2021) and the EU (EU Pesticides Database), five concentrations (0.01, 0.05, 0.5, 10, and 100 mg/kg for thiacloprid and 0.01, 0.05, 0.5, 50, and 100 mg/kg for tolfenpyrad) were added to the various tea samples to determine the average recovery.

For the samples that added more than 10 mg/kg pesticides, they were diluted 10 times with corresponding blank matrix solutions before the UHPLC–MS/MS analysis due to the high sensitivity of the instrument [32]. The limit of quantification (LOQ) was defined as the lowest concentration validated with an acceptable recovery of $\geq$70% and the relative standard deviation (RSD) lower than 20%.

*2.6. Dissipation Dynamics of Thiacloprid and Tolfenpyrad*

The dissipation kinetics of thiacloprid and tolfenpyrad in fresh or dry tea were evaluated by the single first-order kinetics Equation (2), which promoted the evaluation of the half-life ($T_{1/2}$) of the two pesticides, according to the Equation (3):

$$C = C_0 e^{-kt} \tag{2}$$

$$T_{1/2} = \ln 2/k \tag{3}$$

Here, C is the residue concentration (mg/kg) of the pesticide at time point t (day). $C_0$ is the initial concentration (mg/kg) of the pesticide and k is the degradation rate.

*2.7. Dietary Risk Assessment*

For the safety application of thiacloprid and tolfenpyrad, the hazard quotient (HQ) model and risk quotient (RQ) model were applied. The HQ model can evaluate the acute dietary risk assessment by Equations (4) and (5):

$$NESTI = LP \times HR/bw \tag{4}$$

$$RQ_a\% = NESTI/ARfD \times 100\% \tag{5}$$

In these two equations, NESTI (mg/kg bw) refers to the national estimated daily intake, and LP (kg) is the highest large portion reported of food per day. HR (mg/kg) is the highest residue in field trails. As for the bw, it is regarded as the average body weight of a Chinese adult (63 kg). ARfD (mg/kg bw) refers to the acute reference dose.

When it comes to the RQ model, it can help the evaluation of the chronic dietary risk assessment of the two pesticides expressed as $RQ_c$, which can be calculated by Equations (6) and (7):

$$NEDI = [\Sigma(STMRi \times Fi)]/bw \tag{6}$$

$$RQ_c\% = NEDI/ADI \times 100\% \tag{7}$$

NEDI (mg/kg bw) refers to the national estimated daily intake, ADI (mg/kg bw) is the acceptable daily intake, STMRi (mg/kg) refers to the supervised trials median residue (mg/kg), and Fi is the consumption of crop foods (g/d). Risk correlates with the RQ value. An RQ of $\leq$100% implies that a chronic risk to humans is acceptable, whereas an RQ of >100% represents an unacceptable risk.

## 3. Results and Discussion

### 3.1. Validation of the Method

#### 3.1.1. Matrix Effect, Linearity, and LOQ Values

In developing the method, we evaluated the matrix effect by comparing the detector response for each standard solution of thiacloprid and tolfenpyrad with that for each fresh and dry tea extract into which the two pesticides were added at different concentrations. Based on the results, corresponding values for peak area were calculated by measuring the analyte concentration in the different standard working solutions with six concentrations. According to the peak area, a linear regression equation was established. Table 1 shows that the linear relationship between the concentration and peak area was acceptable (i.e., $R^2 > 0.99$) over the range of seven concentrations. However, the matrix effects caused by thiacloprid and tolfenpyrad were noticed, especially for the effect of thiacloprid on fresh tea, whose value reached $-69$%. However, when thiacloprid was applied in other matrixes,

such as cowpeas and citrus, it also produced relatively significant matrix effects [33,34]. Therefore, to maximize the accuracy of the method, the concentration ranges for thiacloprid and tolfenpyrad in the matrices of fresh and dry tea were chosen to derive standard curves, and 0.01 mg/kg was regarded as the LOQ value for each of the pesticides in these matrices.

**Table 1.** Comparison of matrix-matched and solvent calibration of thiacloprid and tolfenpyrad.

| Pesticide | Linear Range (mg/L) | Matrix | Regression Equation | $R^2$ | ME (%) | LOQ (mg/kg) |
|---|---|---|---|---|---|---|
| Thiacloprid | | Solvent | y = 246,672.429762 x + 701.808469 | 0.99074 | – | – |
| | | Fresh tea | y = 76,095.679815 x + 40.610964 | 0.99975 | −69 | 0.01 |
| | 0.001–1.2 | Dry tea | y = 182,759.029991 x + 310.853573 | 0.99567 | −26 | 0.01 |
| Tolfenpyrad | | Solvent | y = 401,114.462560 x + 141.779780 | 0.99985 | – | – |
| | | Fresh tea | y = 350,454.841273 x + 64.389752 | 0.99952 | −13 | 0.01 |
| | | Dry tea | y = 425,190.369489 x − 152.466303 | 0.99684 | 6 | 0.01 |

### 3.1.2. Accuracy and Precision

To determine the accuracy and precision of the method, we performed recovery experiments. Samples of fresh tea and dry tea were spiked with five concentrations of thiacloprid and tolfenpyrad. These parameters were reported as the mean recovery (%) and relative standard deviation (RSD) for these two pesticides for each spiked sample using external matrix-matched standards for quantification. As shown in Table 2, the mean recovery of thiacloprid in fresh tea samples ranged from 73% to 90% and the RSD values ranged from 4.0% to 8.3%. Additionally, dry tea recoveries ranged from 78% to 97%, while their RSD values ranged from 0.8% to 6.6%. For tolfenpyrad, the recovery on fresh tea samples ranged from 75% to 95%, and RSDs ranged from 0.7% to 5.1%. Recovery ranged from 92% to 105% among samples of dry tea, while RSDs ranged from 1.7% to 7.3%. All values for average recovery and RSD were consistent with the standards of SANTE/2021/12830, which requires a recovery range of 70% to 110% and RSD values of less than 10%. Therefore, our method could be suitable for the analysis of thiacloprid and tolfenpyrad in tea samples collected from fields and industries.

**Table 2.** The mean recovery (%) and RSD (%) for thiacloprid and tolfenpyrad from samples of dry and fresh tea at five spiked concentrations (*n* = 5).

| Spiked Concentrations (mg/kg) | Thiacloprid | | | | Tolfenpyrad | | | |
|---|---|---|---|---|---|---|---|---|
| | Fresh Tea | | Dry Tea | | Fresh Tea | | Dry Tea | |
| | Recovery | RSD | Recovery | RSD | Recovery | RSD | Recovery | RSD |
| 0.01 | 73 | 4.0 | 78 | 6.6 | 95 | 5.1 | 105 | 7.3 |
| 0.05 | 88 | 8.3 | 92 | 0.8 | 75 | 3.4 | 93 | 2.1 |
| 0.5 | 90 | 5.4 | 97 | 1.2 | 79 | 3.1 | 92 | 6.1 |
| 10 | 82 | 5.5 | 92 | 2.0 | – | – | – | – |
| 50 | – | – | – | – | 86 | 0.7 | 93 | 1.7 |
| 100 | 82 | 4.6 | 94 | 2.7 | 79 | 1.8 | 99 | 5.1 |

### 3.2. Dissipation Dynamics of Thiacloprid and Tolfenpyrad

After applying the two pesticides, the residue levels of fresh samples and dry samples were measured. In this study, the initial residues in fresh tea leaves during the day 0 (2 h) ranged from 4.5–12.0 mg/kg for thiacloprid and 2.8–6.6 mg/kg for tolfenpyrad, and the range of dry tea was 1.8–4.5 mg/kg for thiacloprid and 7.2–23.0 mg/kg for tolfenpyrad. After 28 days, the residue levels in fresh tea samples gradually decreased by more than 84.6% for thiacloprid and 76.2% for tolfenpyrad. The reducing effects were more significant in dry tea samples, which means that the decrease in thiacloprid and tolfenpyrad was more than 95.1% and 94.4%, respectively. At last, the dissipation of thiacloprid and tolfenpyrad followed the first-order kinetics, with the $R^2$ values ranging from 0.8126 to 0.9950. As

shown in Table 3, the half-lives of thiacloprid were slightly shorter than those of tolfenpyrad. Specifically, the half-lives in fresh tea leaves were in the range of 3.22–8.45 days for thiacloprid and 2.91–10.50 days for tolfenpyrad. For dry tea samples collected from four places, the values of half-lives ranged from 4.72 to 6.48 days for thiacloprid, as well as from 3.67 to 6.73 days for tolfenpyrad. As for the half-life differences of different regions, we can find that the half-lives of the two pesticides in Hubei province were the longest compared with other places. Table 3 also showed that the values of Fujian province were lower than those in other locations. The differences in their half-lives may be caused by the varieties of tea, the local environment, and growth status [29]. Compared with the dissipation of thiacloprid in Asian pear, the half-lives of tea were similar [13]. However, the dissipation of tolfenpyrad in tea was faster than it in citrus (13.3–28.9 days) [34]. In a word, thiacloprid and tolfenpyrad applied in tea leaves could be easily dissipated.

**Table 3.** The residue levels and relative dissipation parameters of thiacloprid and tolfenpyrad in tea were collected at different times.

| Location | Collecting Time | Dry Tea | | | | Fresh Tea | | | |
|---|---|---|---|---|---|---|---|---|---|
| | | Thiacloprid (mg/kg) | Degradation Rate (%) | Tolfenpyrad (mg/kg) | Degradation Rate (%) | Thiacloprid (mg/kg) | Degradation Rate (%) | Tolfenpyrad (mg/kg) | Degradation Rate (%) |
| Anhui | 0 d | 1.815 ± 0.027 | - | 7.963 ± 0.013 | - | 6.359 ± 0.350 | - | 3.266 ± 0.084 | - |
| | 7 d | 0.205 ± 0.001 | 88.7 | 1.534 ± 0.027 | 80.7 | 0.444 ± 0.002 | 93.0 | 0.341 ± 0.009 | 89.6 |
| | 14 d | 0.175 ± 0.002 | 90.4 | 0.684 ± 0.002 | 91.4 | 0.356 ± 0.009 | 94.4 | 0.121 ± 0.006 | 96.3 |
| | 21 d | 0.179 ± 0.005 | 90.1 | 0.218 ± 0.018 | 97.3 | 0.053 ± 0.001 | 99.2 | 0.010 ± 0.001 | 99.7 |
| | 28 d | 0.015 ± 0.001 | 99.2 | 0.033 ± 0.001 | 99.6 | 0.010 ± 0.001 | 99.8 | <0.01 [a] | >99.7 |
| | Dissipation kinetic equations | $C_t = 1.2333e^{-0.138t}$ | | $C_t = 7.5532e^{-0.184t}$ | | $C_t = 4.4789e^{-0.215t}$ | | $C_t = 2.5008e^{-0.236t}$ | |
| | $R^2$ | 0.9136 | | 0.9915 | | 0.9749 | | 0.9920 | |
| | Half-lives (days) | 5.02 | | 3.76 | | 3.22 | | 2.94 | |
| Hubei | 0 d | 3.374 ± 0.002 | - | 15.977 ± 0.234 | - | 4.535 ± 0.022 | - | 2.792 ± 0.074 | - |
| | 7 d | 0.701 ± 0.018 | 79.2 | 9.412 ± 0.088 | 41.1 | 2.289 ± 0.099 | 49.5 | 2.255 ± 0.020 | 19.2 |
| | 14 d | 0.334 ± 0.002 | 90.1 | 3.997 ± 0.092 | 75.0 | 0.444 ± 0.005 | 90.2 | 0.266 ± 0.010 | 90.5 |
| | 21 d | 0.162 ± 0.001 | 95.2 | 2.292 ± 0.051 | 85.6 | 0.317 ± 0.023 | 93.0 | 0.401 ± 0.017 | 85.6 |
| | 28 d | 0.164 ± 0.008 | 95.1 | 0.890 ± 0.035 | 94.4 | 0.701 ± 0.003 | 84.6 | 0.664 ± 0.002 | 76.2 |
| | Dissipation kinetic equations | $C_t = 2.0766e^{-0.107t}$ | | $C_t = 17.459e^{-0.103t}$ | | $C_t = 3.1489e^{-0.082t}$ | | $C_t = 2.1347e^{-0.066t}$ | |
| | $R^2$ | 0.9271 | | 0.9879 | | 0.9312 | | 0.8126 | |
| | Half-lives (days) | 6.48 | | 6.73 | | 8.45 | | 10.50 | |
| Guangxi | 0 d | 4.467 ± 0.080 | - | 23.032 ± 0.059 | - | 12.049 ± 0.207 | - | 6.623 ± 0.002 | - |
| | 7 d | 0.316 ± 0.004 | 92.9 | 2.954 ± 0.039 | 87.2 | 0.932 ± 0.040 | 92.3 | 0.596 ± 0.017 | 91.0 |
| | 14 d | 0.182 ± 0.003 | 95.9 | 0.986 ± 0.020 | 95.7 | 0.569 ± 0.003 | 95.3 | 0.136 ± 0.001 | 97.9 |
| | 21 d | 0.074 ± 0.001 | 98.3 | 0.299 ± 0.008 | 98.7 | 0.527 ± 0.041 | 95.6 | 0.103 ± 0.002 | 98.4 |
| | 28 d | 0.059 ± 0.002 | 98.7 | 0.176 ± 0.003 | 99.2 | 0.203 ± 0.001 | 98.3 | 0.030 ± 0.001 | 99.5 |
| | Dissipation kinetic equations | $C_t = 1.9412e^{-0.144t}$ | | $C_t = 14.302e^{-0.172t}$ | | $C_t = 5.3193e^{-0.129t}$ | | $C_t = 3.4155e^{-0.179t}$ | |
| | $R^2$ | 0.9198 | | 0.9716 | | 0.8892 | | 0.9634 | |
| | Half-lives (days) | 4.81 | | 4.03 | | 5.54 | | 3.87 | |
| Fujian | 0 d | 1.921 ± 0.003 | - | 7.292 ± 0.083 | - | 6.595 ± 0.156 | - | 3.358 ± 0.162 | - |
| | 7 d | 0.239 ± 0.007 | 87.6 | 1.371 ± 0.003 | 81.2 | 0.757 ± 0.052 | 88.5 | 0.387 ± 0.006 | 88.5 |
| | 14 d | 0.069 ± 0.001 | 96.4 | 0.196 ± 0.005 | 97.3 | 0.103 ± 0.009 | 98.4 | 0.021 ± 0.001 | 99.4 |
| | 21 d | 0.042 ± 0.001 | 97.8 | 0.078 ± 0.001 | 98.9 | 0.076 ± 0.002 | 98.9 | 0.010 ± 0.001 | 99.7 |
| | 28 d | 0.026 ± 0.001 | 98.6 | 0.041 ± 0.002 | 99.4 | 0.025 ± 0.001 | 99.6 | <0.01 [a] | >99.7 |
| | Dissipation kinetic equations | $C_t = 1.0103e^{-0.147t}$ | | $C_t = 5.1161e^{-0.189t}$ | | $C_t = 3.6697e^{-0.192t}$ | | $C_t = 1.8904e^{-0.238t}$ | |
| | $R^2$ | 0.9479 | | 0.9938 | | 0.9794 | | 0.9950 | |
| | Half-lives (days) | 4.72 | | 3.67 | | 3.61 | | 2.91 | |

[a]: <0.01 represents the residue was lower than the LOQ value, and the half value of LOQ (0.005 mg/kg) was taken to fit the dissipation kinetic equations if the residue was lower than the LOQ.

### 3.3. Terminal Residues of Thiacloprid and Tolfenpyrad in Tea

After applying the two pesticides, the residues were measured at two pre-harvest intervals (PHIs) and represented in Table 4. For thiacloprid, the residues in fresh tea were higher than it in dry tea. It shows that the residue levels were in the ranges of 0.04–2.55 mg/kg for fresh samples and 0.04–1.23 mg/kg for dry samples. This may indicate that a high-temperature process can promote the degradation of thiacloprid, which showed a similar degradation pattern of thiacloprid through the production of *Lonicera japonica* [35]. However, the residues of tolfenpyrad showed an opposite trend compared with the level trend of thiacloprid. The final residue levels were in ranges of <0.01–1.38 mg/kg for fresh tea and 0.08–4.00 mg/kg for dry tea. Since some processing factors of the production steps of dry tea are higher than one, the residual levels of tolfenpyrad may be condensed during the manufacturing [28]. All of these proved that tea processing plays an important role in the changes in pesticide levels. In addition, geographical conditions played an essential role in the residues of these two pesticides. The residue levels of the two pesticides in Fujian province and Guangdong province were lower than the levels of most of the other places (Table 4). Compared with these collecting locations, Guangdong province and Fujian province are near the sea in the south of China, whose temperature and precipitation are higher than other locations. Therefore, the low residual levels may be related to these conditions, which was consistent with previous studies [36,37].

**Table 4.** Terminal residues of thiacloprid and tolfenpyrad in different tea matrices at different preharvest intervals (PHIs).

| Location | Matrix | 14 d | | 21 d | | Matrix | 14 d | | 21 d | |
| | | Thiacloprid (mg/kg) | Tolfenpyrad (mg/kg) | Thiacloprid (mg/kg) | Tolfenpyrad (mg/kg) | | Thiacloprid (mg/kg) | Tolfenpyrad (mg/kg) | Thiacloprid (mg/kg) | Tolfenpyrad (mg/kg) |
|---|---|---|---|---|---|---|---|---|---|---|
| Anhui | | 0.175 ± 0.002 | 0.684 ± 0.002 | 0.179 ± 0.005 | 0.218 ± 0.018 | | 0.356 ± 0.009 | 0.121 ± 0.006 | 0.053 ± 0.001 | 0.010 ± 0.001 |
| Hubei | | 0.334 ± 0.002 | 3.997 ± 0.092 | 0.162 ± 0.001 | 2.292 ± 0.051 | | 0.444 ± 0.005 | 0.266 ± 0.010 | 0.317 ± 0.023 | 0.401 ± 0.010 |
| Guangxi | | 0.182 ± 0.003 | 0.986 ± 0.020 | 0.074 ± 0.001 | 0.299 ± 0.008 | | 0.569 ± 0.003 | 0.136 ± 0.001 | 0.527 ± 0.041 | 0.103 ± 0.002 |
| Fujian | | 0.069 ± 0.001 | 0.196 ± 0.005 | 0.042 ± 0.001 | 0.078 ± 0.001 | | 0.103 ± 0.009 | 0.021 ± 0.001 | 0.076 ± 0.002 | 0.010 ± 0.001 |
| Shandong | Dry | 0.360 ± 0.010 | 1.893 ± 0.007 | 0.208 ± 0.008 | 0.478 ± 0.003 | Fresh | 0.518 ± 0.007 | 0.078 ± 0.006 | 0.038 ± 0.008 | <0.01 [a] |
| Zhejiang | tea | 1.226 ± 0.009 | 1.678 ± 0.005 | 0.927 ± 0.009 | 0.315 ± 0.009 | tea | 1.304 ± 0.016 | 1.379 ± 0.011 | 1.104 ± 0.007 | 0.215 ± 0.006 |
| Hunan | | 0.944 ± 0.019 | 1.899 ± 0.051 | 0.797 ± 0.011 | 2.347 ± 0.038 | | 2.547 ± 0.070 | 0.576 ± 0.008 | 2.549 ± 0.014 | 0.457 ± 0.017 |
| Guizhou | | 0.104 ± 0.007 | 0.282 ± 0.001 | 0.054 ± 0.001 | 0.080 ± 0.005 | | 0.309 ± 0.009 | 0.050 ± 0.001 | 0.203 ± 0.014 | 0.017 ± 0.002 |
| Yunnan | | 0.687 ± 0.023 | 3.367 ± 0.014 | 0.416 ± 0.001 | 0.497 ± 0.002 | | 1.098 ± 0.020 | 1.204 ± 0.021 | 0.702 ± 0.009 | 0.459 ± 0.018 |
| Guangdong | | 0.094 ± 0.008 | 0.276 ± 0.004 | 0.059 ± 0.006 | 0.075 ± 0.002 | | 0.359 ± 0.004 | 0.054 ± 0.001 | 0.159 ± 0.008 | 0.018 ± 0.004 |

[a]: <0.01 represents the residue was lower than the LOQ value.

To choose the proper PHI to develop the guideline of thiacloprid and tolfenpyrad, we focused on the residues of dry tea samples. For thiacloprid, its residue at the interval of 14 days or 21 days was greatly lower than the maximum residue level (MRL) measured in tea (10 mg/kg in the EU, China, and the CAC; 25 mg/kg in Japan). For tolfenpyrad, the residue levels of tolfenpyrad from two intervals were greatly lower than the MRL values from diverse regions (30 mg/kg in Japan and Korea; 50 mg/kg in China). Therefore, 14 days is suggested as a safe PHI for thiacloprid and tolfenpyrad in tea. However, the residue level of tolfenpyrad was higher than the MRL required by the CAC (0.05 mg/kg) and the EU (0.01 mg/kg), which means the pesticide may influence the export of the tea in some regions.

### 3.4. Dietary Exposure Risk Assessment

Considering the primary data for dry tea, we estimated the acute and chronic dietary risk assessments of thiacloprid and tolfenpyrad. The calculation process is shown in Supplementary Tables S1–S5. For the chronic risk of the two pesticides on tea, ARfD values were obtained from the Joint Meeting on Pesticide Residues (JMPR) official reports [38,39]. We calculated the $RQ_a$ values (Table 5) of two different intervals, which were in the range of 3.74–4.94% for thiacloprid and 28.3–48.2% for tolfenpyrad. Since all results were below 100%, the acute potential risk of the two pesticides could be acceptable in tea.

**Table 5.** Dietary risk assessment of thiacloprid and tolfenpyrad in tea at two different intervals.

| Pesticide | PHI (Day) | ARfD (mg/kg bw) | ADI (mg/kg bw) | STMR (mg/kg) | HR (mg/kg) | $RQ_a$ (%) | $RQ_c$ (%) |
|---|---|---|---|---|---|---|---|
| Thiacloprid | 14 | 0.03 | 0.01 | 0.26 | 1.23 | 4.94 | 47.4 |
| | 21 | | | 0.17 | 0.93 | 3.74 | 47.2 |
| Tolfenpyrad | 14 | 0.01 | 0.006 | 1.34 | 4.00 | 48.2 | 28.6 |
| | 21 | | | 0.31 | 2.35 | 28.3 | 25.3 |

Since the RQ model takes the whole diet into account, $RQ_c$ is a more realistic and comprehensive index to measure the exposure risk of thiacloprid and tolfenpyrad. Comparing MRLs of the pesticides on registered crops in China, higher MRLs for each classification in Chinese dietary pattern were determined. As for the classification containing the crop of tea, STMR values were chosen to calculate NEDI values. All of these can help the evaluated risk be closer to reality. As presented in Table 5, the ranges of STMR were 0.17–0.26 mg/kg for thiacloprid and 0.31–1.34 mg/kg for tolfenpyrad, respectively. Then, $RQ_c$ values were calculated. For thiacloprid, its risk quotients of two intervals were from 47.2% to 47.4%, and the $RQ_c$ values were in the range of 25.3–28.6% for tolfenpyrad. Since all values of $RQ_c$ were lower than 100%, the chronic potential risk posed by the tea applied with the two pesticides can be accepted by the general population.

## 4. Conclusions

A quick and reliable method for detecting thiacloprid and tolfenpyrad residues in tea with high accuracy and precision was developed. The LOQ for the two compounds was 0.01 mg/kg. The half-lives were in ranges of 3.22–8.45 days for thiacloprid and 2.91–10.50 days for tolfenpyrad. The terminal residue levels of fresh tea were higher than those of dry tea for thiacloprid. However, the trend of terminal residues of tolfenpyrad represented was the opposite. According to those data, we suggested 14 days as the safe preharvest interval for applying thiacloprid and tolfenpyrad in tea. Based on the safety risk assessments using the residue data, the $RQ_a$ and $RQ_c$ values of thiacloprid and tolfenpyrad were all less than 100%, implying an acceptable risk for consumers. This study lays the foundation for the residual dynamics of thiacloprid and tolfenpyrad. It also evaluates their dietary risk for the safe application of a mixture of the two pesticides in tea.

**Supplementary Materials:** The following supporting information can be downloaded at https://www.mdpi.com/article/10.3390/agronomy12123166/s1, Table S1: The acceptable daily intake (ADI, mg/kg body weight), Chinese dietary pattern, supervised trials and median residues (STMR, mg/kg), the corresponding MRLs registered, and risk quotient (RQ, %) of thiacloprid in dry tea when the PHI was 14 days; Table S2: The acceptable daily intake (ADI, mg/kg body weight), Chinese dietary pattern, supervised trials and median residues (STMR, mg/kg), the corresponding MRLs registered, and risk quotient (RQ, %) of thiacloprid in dry tea when the PHI was 21 days; Table S3: The acceptable daily intake (ADI, mg/kg body weight), Chinese dietary pattern, supervised trials and median residues (STMR, mg/kg), the corresponding MRLs registered, and risk quotient (RQ, %) of tolfenpyrad in dry tea when the PHI was 14 days; Table S4 The acceptable daily intake (ADI, mg/kg body weight), Chinese dietary pattern, supervised trials and median residues (STMR, mg/kg), the corresponding MRLs registered, and risk quotient (RQ, %) of tolfenpyrad in dry tea when the PHI was 21 days; Table S5: Acute dietary risk assessment of thiacloprid and tolfenpyrad in tea.

**Author Contributions:** Conceptualization, Y.Z.; methodology, W.W.; validation, D.G., J.L., and H.L.; formal analysis, H.C.; investigation, H.C.; resources, Y.Z.; writing—original draft preparation, W.W.; writing—review and editing, Y.Z. and H.C. All authors have read and agreed to the published version of the manuscript.

**Funding:** This research was funded by the Open Fund Project of the Key Laboratory of Tea Quality and Safety Control, Ministry of Agriculture and Rural Affairs, TQSC202203.

**Data Availability Statement:** Not applicable.

**Conflicts of Interest:** The authors declare no conflict of interest.

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
