# Peer review of "Dissipation and Dietary Risk Assessment of Thiacloprid and Tolfenpyrad in Tea in China"

_agronomy, doi:10.3390/agronomy12123166_

Round 1

Reviewer 1 Report

The manuscript describes an investigation of the dissipation of thiacloprid and tolfenpyrad in tea plants and conducting the dietary risk assessment of the 2 pesticides.  The dissipation study was conducted in at least 12 different locations in China (Hefei City, Anhui Province; Qingyuan City, Guangdong Province; Nanning City, Guangxi Province; Guiyang City, Guizhou Province; Enshi Tujia and Miao Autonomous Prefecture, Hubei Province; Changsha City, Hunan Province; Ningbo City, Zhejiang Province; Taian City, Shandong Province; Fu‘an City, Fujian Province; Kunming City, Yunnan Province).  The authors only show some of the results obtained from only 4 locations (Anhui, Hubei, Guangxi, and Fujian).  In addition, although sampling was conducted at 5 time intervals, the authors only list the results of the initial, PHI 14 and 21 days. The following actions are recommendations to ensure all data obtained are reported:

1.      A map showing the locations of all sampling sites.  Readers are not familiar with Chinese cities and their locations.

2.      List the results of samples collected from all locations (at least 12)

3.      List the results of all time intervals (0, 7, 14, 21, 28 days PHI).

4.      A discussion about the residues found in different geographic locations, i.e. the effect of climate/elevation, etc., on the amounts of residues at sampling times.

In addition, the following comments should be addressed:

1.      In general, the authors always refer to the 2 studied pesticides as “two pesticides” throughout the manuscript, or two substances (in the abstract).  This should be replaced by “the two pesticides” or “thiacloprid and tolfenpyrad”

2.      In the abstract: “However, there is scarce information on their dissipation and dietary risk, which poses a threat to human health” is an inaccurate statement and should be removed.  There are numerous studies on the dissipation and dietary risk of pesticides conducted all over the world.

3.      In the Introduction: Line 26: State the common name of the type of tea investigated, such as black tea, green tea,….

4.      Line 29: delete “substances” or replace with” aromatic substances”

5.      Line 30:  replace “pharmacological” with “health”

6.      Line 31: “can against several diseases” can what?  protect?

7.      Line 38: “a highly economic way”.  The meaning is not understood.

8.      Line 40:  replace “toxicological” with “toxic”

9.      Line 50: “replace “effective against to the pests resisting to pesticides …” with “effective against pests resisting pesticides…”

10.  Line 63:  “focused on the thiacloprid or tolfenpyrad…” is not understood. 

11.  In Field Trials:  Line 108:  Experimental area should be plural (areas…..were sprayed…) or if only one area say “the experimental area”

12.  Line 110:  replace “random” with “randomly”

13.  In Method Validation: Line 156: “by China and EU” state the reference and Agency that set the MRLs

14.  In the Results: Line 194: “… the linear relationship between the injection volume and the peak area…”  replace with: “… the linear relationship between the concentration (or amount) and peak area…”.  The injection volume is always constant at 1 mL as indicated in the Materials and Methods.

15.  Line 206 and title of Table 2: “spiked with four concentrations…”  this should be five concentrations as stated in Materials and Methods.

16.  Line 225: “With increasing time…”  State the exact time (28 days?)

17.  The structure of Table 4 should be changed.  It is not understood what all the number represent under “Residue (mg/kg)”

Author Response

Dear Reviewer,

We greatly appreciate you reviewing our manuscript (agronomy-2050198) carefully and giving us so many helpful suggestions. Based on these comments and advice, we have made careful modifications to the manuscript. The main corrections in the paper and the responses to your comments are as follows:

Decision letter:

The manuscript describes an investigation of the dissipation of thiacloprid and tolfenpyrad in tea plants and conducting the dietary risk assessment of the 2 pesticides. The dissipation study was conducted in at least 12 different locations in China (Hefei City, Anhui Province; Qingyuan City, Guangdong Province; Nanning City, Guangxi Province; Guiyang City, Guizhou Province; Enshi Tujia and Miao Autonomous Prefecture, Hubei Province; Changsha City, Hunan Province; Ningbo City, Zhejiang Province; Taian City, Shandong Province; Fu‘an City, Fujian Province; Kunming City, Yunnan Province). The authors only show some of the results obtained from only 4 locations (Anhui, Hubei, Guangxi, and Fujian). In addition, although sampling was conducted at 5 time intervals, the authors only list the results of the initial, PHI 14 and 21 days. The following actions are recommendations to ensure all data obtained are reported:

  1. A map showing the locations of all sampling sites. Readers are not familiar with Chinese cities and their locations.

A: Thanks for your suggestion. We have added the Figure. 2 (after Line 118) to show all locations of all sampling sites, which can help readers be familiar with Chinese cities and relative locations.

  1. List the results of samples collected from all locations (at least 12)

A: Thank you so much. We have changed it in Table 4.

  1. List the results of all time intervals (0, 7, 14, 21, 28 days PHI).

A: We greatly appreciate your advice. In order to bring the direct feelings to the readers, we added the results of all time intervals in Table 3.

  1. A discussion about the residues found in different geographic locations, i.e. the effect of climate/elevation, etc., on the amounts of residues at sampling times.

A: Thanks for your great comment. We have discussed more in the sections 3.2 and 3.3.

In addition, the following comments should be addressed:

  1. In general, the authors always refer to the 2 studied pesticides as “two pesticides” throughout the manuscript, or two substances (in the abstract). This should be replaced by “the two pesticides” or “thiacloprid and tolfenpyrad”

A: Thanks for your advice. We have replaced “substances” by “two pesticides” or “thiacloprid and tolfenpyrad”. Please refer line 13, 14, 16, 20, 146, 174, 186, 198, 204, 294, 298, 310 and 324.

  1. In the abstract: “However, there is scarce information on their dissipation and dietary risk, which poses a threat to human health” is an inaccurate statement and should be removed. There are numerous studies on the dissipation and dietary risk of pesticides conducted all over the world.

A: We greatly appreciate your comments. And we changed another expression to “For achieving satisfactory effects on pests, multiple pesticides in a low application dose have been the trend at present”. Please refer Line 10 to 11.

  1. In the Introduction: Line 26: State the common name of the type of tea investigated, such as black tea, green tea,….

A: Thank you so much. We have specified the tea as green tea in Line 26.

  1. Line 29: delete “substances” or replace with” aromatic substances”

A: Thanks a lot. We have deleted “substances” in Line 29.

  1. Line 30: replace “pharmacological” with “health”

A: Thanks a lot. We have corrected it in Line 29, which readers are familiar with.

  1. Line 31: “can against several diseases” can what? protect?

A: Thank you so much. We have corrected the wrong places by adding “protect” in Line 30.

  1. Line 38: “a highly economic way”. The meaning is not understood.

A: Thanks for your advice. We just want to express that pesticide is a simple method to control the effects of pests with a low cost. And we changed the expression in Line 39.

  1. Line 40: replace “toxicological” with “toxic”

A: Thank you so much. We have replaced the wrong expression by toxic in Line 42.

  1. Line 50: “replace “effective against to the pests resisting to pesticides …” with “effective against pests resisting pesticides…”

A: Thanks for your great advice. We have corrected it in Line 52 which greatly made the sentence concise.

  1. Line 63: “focused on the thiacloprid or tolfenpyrad…” is not understood. 

A: Thanks for your comment. What we want express is that several studies have focused on either thiacloprid or tolfenpyrad on tea. In order to help readers understand easily, we have changed another expression in Line 65.

  1. In Field Trials: Line 108: Experimental area should be plural (areas…..were sprayed…) or if only one area say “the experimental area”

A: Thanks for your opinion. We have corrected it in Line 112.

  1. Line 110: replace “random” with “randomly”

A: Thank you so much. We have justified the wrong expression by “randomly” in Line 114.

  1. In Method Validation: Line 156: “by China and EU” state the reference and Agency that set the MRLs

A: Thanks a lot. We have noted the reference file (database) in Line 163, which can help readers to find the sources if they do relative work.

  1. In the Results: Line 194: “… the linear relationship between the injection volume and the peak area…” replace with: “… the linear relationship between the concentration (or amount) and peak area…”. The injection volume is always constant at 1 mL as indicated in the Materials and Methods.

A: Thanks a lot. We have changed the expression to “the linear relationship between the concentration and peak area” in Line 202.

  1. Line 206 and title of Table 2: “spiked with four concentrations…” this should be five concentrations as stated in Materials and Methods.

A: We greatly appreciate your opinion. Thank you so much. We have corrected it in Line 229.

  1. Line 225: “With increasing time…” State the exact time (28 days?)

A: Thanks a lot. Actually, we wanted to express the dissipation rate after 28 days, which is same to you. And we changed the expression in Line 234.

  1. The structure of Table 4 should be changed. It is not understood what all the number represent under “Residue (mg/kg)”.

A: Thanks a lot. We have changed Table 4 as you mention in Q4 of part 1. It’s a good suggestion since readers can know more detail about our study.

In a word, thanks for your beneficial and helpful comments. They improved the quality of our manuscript. Thank you so much. If there is any problem with our manuscript, please give us more suggestions to improve it.

Best regards,

Weitao Wang, Dr. Ying Zhang

Reviewer 2 Report

The concentration and dietary risk assessment of thiacloprid and tolfenpyrad of Tea in China were investigated. Analysis method of  two insecticides  was improved.Field trails were conducted in China and residues. And also the dietary risk of  two pesticide was sestimated.The paper is well organized . The data displayed in a concise and easy way to the readership. The data analysis is also comprehensive. However, the novelty of the paper is not so significant, the risk assessment did not have enough data.

Author Response

Dear Reviewer,

Thanks for your recognition and your comments on our manuscript (agronomy-2050198). To be honest, we have compared relative MRL data set by countries like the US, Korea, Japan, and Australia. However, the values they set were different from the standards set by China, which may not be consistent with the current situation in China. Considering the Chinese market and Chinese diet habits, we finally chose the MRL (set by China) for the dietary risk assessment, which can be more accurate for evaluating the risk of the Chinese.

Thanks for taking the time to review our manuscript. If there is any problem with our manuscript, please give us more suggestions to improve it.

Best regards,

Weitao Wang, Dr. Ying Zhang

Reviewer 3 Report

The content of this manuscript is to investigate the dissipation and dietary risk assessment of thiacloprid and tolfenpyrad in tea leaves. For this purpose, a method for the detection of the above compounds was first developed, validated and finally applied to field samples from different locations. Among other things, conclusions for a low-risk application time of the insecticides are derived from this. The methodological work has been carried out very carefully. The results are of great benefit for the estimation of the risk potential. However, a sound scientific discussion taking into account the current state of knowledge is missing. There is also no extra chapter "Discussion". Only in chapter 3 "Results" some results are discussed in a rudimentary way. Further comments on the introduction:

Line 30/31: in the sentence in question, a word between "can" and "against" is missing.

Line 37: what exactly is meant by tea here? The leaves or the whole plant? Please give examples of possible diseases that can affect the tea plant. What exactly is meant by "output"?

Line 62: source for the structural formulas in Figure 1 is missing

Line 77-82: a clear objective is missing; only listed what research was done.

Author Response

Dear Reviewer,

We greatly appreciate you reviewing our manuscript (agronomy-2050198) carefully and giving us some helpful suggestions. Based on these comments and advice, we have made careful modifications to the manuscript. The main corrections in the paper and the responses to your comments are as follows:

Decision letter:

The content of this manuscript is to investigate the dissipation and dietary risk assessment of thiacloprid and tolfenpyrad in tea leaves. For this purpose, a method for the detection of the above compounds was first developed, validated and finally applied to field samples from different locations. Among other things, conclusions for a low-risk application time of the insecticides are derived from this. The methodological work has been carried out very carefully. The results are of great benefit for the estimation of the risk potential. However, a sound scientific discussion taking into account the current state of knowledge is missing. There is also no extra chapter "Discussion". Only in chapter 3 "Results" some results are discussed in a rudimentary way. Further comments on the introduction:

Line 30/31: in the sentence in question, a word between "can" and "against" is missing.

A: Thank you so much. We have added “protect” in Line 30.

Line 37: what exactly is meant by tea here? The leaves or the whole plant? Please give examples of possible diseases that can affect the tea plant. What exactly is meant by "output"?

A: Thanks a lot. We have corrected the wrong place and added relative discussion from Line 37 to 39. It can help readers know more details why pesticides are important to tea field. And we changed output to another expression referred in Line 40 so that readers can understand easily.

Line 62: source for the structural formulas in Figure 1 is missing

A: Thanks for your comment. The chemical structure is drawn by ChemDraw, which may not need to be cited.

Line 77-82: a clear objective is missing; only listed what research was done.

A: Thank you so much. We have specified our objective in Line 79. It can help readers directly know the reasons why we did this study.

In addition, based on your previous comments, we also discussed more in the sections 3.2 and 3.3, which can help readers know more about our research.

Last but not least, we greatly appreciate you spending time reviewing our manuscript carefully and giving us helpful advice. Because of these, we found that the quality of the manuscript can be improved. Thank you so much. If there is any problem with our manuscript, please give us more suggestions to improve it.

Best regards,

Weitao Wang, Dr. Ying Zhang

Round 2

Reviewer 1 Report

The authors revised the manuscript and addressed most of the comments.  However, the revised manuscript needs the following issues to be addressed:

1. A key (legend) of the province names (at least for the provinces sampled) for the map in Fig. 2 should be provided.

2.  Explain and provide reasons why only the results of 4 provinces are listed in Table 3.  Can results from all sampled sites at each collection time be listed?

3.  In Table 4, the average results of 10 sampling sites are listed.   However, there were 12 sampling sites.  Provide an explanation why they are missing or add the results from the 2 missing sites.

Thank you!

Author Response

Reviewer 1

The authors revised the manuscript and addressed most of the comments.  However, the revised manuscript needs the following issues to be addressed:

  1. A key (legend) of the province names (at least for the provinces sampled) for the map in Fig. 2 should be provided.

A: We greatly appreciate your comment. We have added more information in Figure. 2 (in Line 119). In this way, we believe readers can know more detail about the sites of our collecting locations.

  1. Explain and provide reasons why only the results of 4 provinces are listed in Table 3. Can results from all sampled sites at each collection time be listed?

A: Thanks for your question. Based on the standard NY/T 788-2018 issued by the Ministry of Agriculture and Rural Affairs of China [1], it requests that if the number of experimental locations is more than eight, the dissipation data should be collected in at least four of these locations (specifically in section 7.6 of the standard). For Table 3, it described the dissipation of thiacloprid and tolfenpyrad. And Table 4 just described the terminal residues of the two pesticides. Since our experiment was conducted in ten locations in China, we provided the dissipation data from Anhui province, Hubei province, Guangxi province and Fujian province.

[1] Ministry of Agriculture and Rural Affairs of the People’s Republic of China. (2018). NY/T 788-2018 Guideline for the Testing of Pesticide Residues in Crops.

  1. In Table 4, the average results of 10 sampling sites are listed.  However, there were 12 sampling sites.  Provide an explanation why they are missing or add the results from the 2 missing sites.

A: Thanks for your comment. We are so sorry that our expression made you feel confused. As we mentioned in Q2, we conducted our experiment in 10 locations. And for each location, the samples were collected randomly from 12 sampling points in the experimental area. Therefore, in order not to make the readers confused, we have changed some expressions in Section 2.2 (Specifically in Line 104 and 114).

These are all the responses to your comments. Thanks for your careful review. Thank you so much.

Reviewer 2 Report

The manuscript was revised according the comments. However, Fig 2(The map of locations of all sampling sites), Chinese Map is serious mistake, which is not allowed by Chinese law.

Author Response

Reviewer 2

The manuscript was revised according to the comments. However, Fig 2 (The map of locations of all sampling sites), Chinese Map is serious mistake, which is not allowed by Chinese law.

A: We are incredibly grateful that you helped us find the mistake. And we have changed Figure 2 (in Line 119). Great thanks for your advice.

If there is any problem with our manuscript, please feel free to contact us. Thanks for your careful review. Thank you so much.

Reviewer 3 Report

The title of chapter 3 is Results. Since the revised version of the manuscript also includes a discussion of the results, this should also be evident in the title of the chapter.

All other comments were considered.

Author Response

Reviewer 3

The title of chapter 3 is Results. Since the revised version of the manuscript also includes a discussion of the results, this should also be evident in the title of the chapter.

All other comments were considered.

A: Thanks for your comment. We have already added “and Discussion” in the title of Chapter 3 in Line 193.

Thanks for your careful review. Thank you so much.